# Acss2 Deletion Reveals Functional Versatility via Tissue-Specific Roles in Transcriptional Regulation

**DOI:** 10.3390/ijms24043673

**Published:** 2023-02-12

**Authors:** Narayanan Puthillathu Vasudevan, Dharmendra K. Soni, John R. Moffett, Jishnu K. S. Krishnan, Abhilash P. Appu, Sarani Ghoshal, Peethambaran Arun, John M. Denu, Thomas P. Flagg, Roopa Biswas, Aryan M. Namboodiri

**Affiliations:** 1Department of Anatomy, Physiology, and Genetics, Uniformed Services University of the Health Sciences, Bethesda, MD 20814, USA; 2Department of Biomolecular Chemistry, University of Wisconsin-Madison, Madison, WI 53706, USA; 3Wisconsin Institute for Discovery, University of Wisconsin-Madison, Madison, WI 53706, USA

**Keywords:** acetate, acetyl-coenzyme A, ATP citrate lyase, acetyl-CoA synthetase, acetylation, transcriptional regulation

## Abstract

The coordination of cellular biological processes is regulated in part via metabolic enzymes acting to match cellular metabolism to current conditions. The acetate activating enzyme, acyl-coenzyme A synthetase short-chain family member 2 (Acss2), has long been considered to have a predominantly lipogenic function. More recent evidence suggests that this enzyme has regulatory functions in addition to its role in providing acetyl-CoA for lipid synthesis. We used Acss2 knockout mice (*Acss2*^−/−^) to further investigate the roles this enzyme plays in three physiologically distinct organ systems that make extensive use of lipid synthesis and storage, including the liver, brain, and adipose tissue. We examined the resulting transcriptomic changes resulting from Acss2 deletion and assessed these changes in relation to fatty acid constitution. We find that loss of Acss2 leads to dysregulation of numerous canonical signaling pathways, upstream transcriptional regulatory molecules, cellular processes, and biological functions, which were distinct in the liver, brain, and mesenteric adipose tissues. The detected organ-specific transcriptional regulatory patterns reflect the complementary functional roles of these organ systems within the context of systemic physiology. While alterations in transcriptional states were evident, the loss of Acss2 resulted in few changes in fatty acid constitution in all three organ systems. Overall, we demonstrate that Acss2 loss institutes organ-specific transcriptional regulatory patterns reflecting the complementary functional roles of these organ systems. Collectively, these findings provide further confirmation that Acss2 regulates key transcription factors and pathways under well-fed, non-stressed conditions and acts as a transcriptional regulatory enzyme.

## 1. Introduction

Biological processes operating within and between intracellular compartments are coordinated, in part, via metabolic enzymes that act as key regulators of cellular metabolism. Certain metabolic enzymes coordinate metabolite sensing and signaling mechanisms to maintain energy homeostasis, including glucose, lipid and amino acid metabolism, energy storage and utilization, and responses to injury and the environment [1,2]. In addition, some of these metabolic enzymes actively shuttle between the cytoplasm and the nucleus enabling them to act as multifunctional regulatory proteins [3,4]. They execute discrete biochemical and signaling roles in the cytoplasm and the nucleus depending upon the needs of cells under varying conditions. Further, these regulatory metabolic enzymes can modulate gene expression through direct or indirect mechanisms whereby they can behave as transcription factors, regulators of transcription factors, or post-translational modifiers of protein function.

Acyl-coenzyme A synthetase short-chain family member 2 (Acss2, also known as acetyl-CoA synthetase-1) is an evolutionarily conserved, lipogenic enzyme [5,6]. It was first identified in rat liver as a cytosolic enzyme that participated in the synthesis of fatty acids and sterols [7,8]. Acss2 catalyzes a reaction that consumes acetate, coenzyme A (CoA) and adenosine triphosphate (ATP) to generate acetyl-CoA, adenosine monophosphate (AMP) and pyrophosphate (enzyme classification; EC 6.2.1.1) [5]. These findings implicate Acss2 as an additional route to cytoplasmic acetyl-CoA generation, parallel to ATP-citrate lyase (Acly). It has been also documented that the expression of Acss2 mRNA is regulated by sterol regulatory element-binding proteins, further substantiating its role in lipid synthesis [9,10,11,12]. Acss2-deficient (*Acss2*^−/−^) mice reproduce well and their young develop normally, indicating that Acss2 is nonessential under normal conditions with adequate food [13]. Thus, Acss2 acts as a parallel pathway to extramitochondrial acetyl-CoA synthesis that is not critical for survival under favorable conditions. Nonetheless, the evolutionary conservation of Acss2 indicates that it provides significant selective advantages.

Despite its clear role in providing acetyl-CoA for lipid synthesis in the cytoplasm, strong Acss2 expression has been observed in the nucleus of most cell types, delineating it as a nucleocytosolic enzyme [14,15,16]. Importantly, Acss2 has been shown to translocate from the cytoplasm to the nucleus in response to nutrient deprivation, stress, or injury [17,18,19]. In the nucleus, Acss2 facilitates acetylation reactions for numerous proteins including histones [20,21]. Acss2 also facilitates acetylation of transcription factors, for example, hypoxia-inducible factor-2 alpha (Hif-2α) [17,22], transcription factor EB (Tfeb) [19] and interferon regulatory factor 4 (Irf4) [23]. The phosphorylation status of Acss2 determines its subcellular localization, with the unphosphorylated form being retained in the cytoplasm and the phosphorylated form being translocated to the nucleus [19]. These studies demonstrate additional regulatory roles of Acss2 in gene transcription, metabolic reprogramming, cell cycle progression, lysosomal biogenesis, and autophagy. The importance of these cellular biological processes in maintaining cellular energy balance, homeostasis, and cell survival is well-known. Therefore, not surprisingly, the dysregulation of Acss2 has been linked to various human diseases, including cardiovascular diseases, cancer, diabetes, and obesity (reviewed in [24,25]).

We performed genome-wide transcriptome profiling (microarray) in normally fed (non-fasted) adult (15–16 weeks old) *Acss2*^−/−^ mice and age-matched wild-type (WT) mice. We examined differentially expressed genes (DEGs) associated with the deletion of Acss2 in metabolically disparate organ systems involved in extensive lipid synthesis, deposition, and mobilization including the liver, brain, and mesenteric adipose tissue. The liver is a site of lipid and ketone body synthesis while adipose tissue is a site of lipid storage and mobilization. The brain is also a site of extensive lipid metabolism and deposition. The data reveal distinct patterns of altered gene expression in the three organ systems examined. Further, analyses of DEGs show that loss of Acss2 leads to dysregulation of numerous canonical signaling pathways, upstream transcriptional regulatory molecules, cellular processes, and biological functions, which were also distinct in the liver, brain, and adipose tissue. Despite such alterations in transcriptional states, cellular pathways, and processes, the analysis of global fatty acid content demonstrated that the loss of Acss2 resulted in few changes in fatty acid constitution in all three organ systems. Our results reveal dynamic reprogramming of gene expression patterns and associated cellular pathways and processes between *Acss2*^−/−^ and WT mice, which were distinct in the liver, brain, and adipose tissue. These findings contribute to the growing knowledge base linking metabolic enzymes such as Acss2 with the dynamic reprogramming of signaling and metabolic pathways necessary to adapt to changing internal and external conditions.

## 2. Results

### 2.1. Fatty Acid Quantification

To determine the effect of loss of Acss2 on fatty acid constitution, we performed liquid chromatography-mass spectrometry (LC-MS) fatty acid analysis in liver, brain, and mesenteric adipose tissues from normally fed (non-fasted) adult (15 to 16 weeks old) *Acss2*^−/−^ (knockout mice, KO) and age-matched WT mice of the same strain. Acss2 deletion did not result in broad alternations in FA levels, but rather there were several specific, significant differences in key fatty acids in a tissue-specific context. Liver eicosanoic acid (a saturated fatty acid; 20:0) was slightly elevated in the knockout mice (KO = 1.55 μg/mL ± 0.18, WT = 1.31 μg/mL ± 0.15, *p* = 0.054). In the brain, oleic acid (a monounsaturated omega-9 fatty acid; 18:1 cis-9) was significantly lower in the KO mice (KO = 1084 μg/mL ± 121, WT = 1401 μg/mL ± 268, *p* = 0.043). Brain stearic acid (a saturated fatty acid; 18:0) was also significantly lower in the KO mice (KO = 773 μg/mL ± 38, WT = 884 μg/mL ± 66, *p* = 0.012). In adipose tissue myristic acid (C14:0) was significantly higher in the KO mice (KO = 22.2 μg/mL ± 3.1, WT = 15.5 μg/mL ± 4.4, *p* = 0.024). No other significant differences were noted (Data in Appendix A). These results indicate that the role of Acss2 in lipogenesis can be mostly compensated for by Acly in normally fed mice, and argues against a strong, selective role for Acss2 in the synthesis of specific fatty acids.

### 2.2. Transcriptional Analysis of Acss2 Deletion

To further investigate the roles of Acss2, we performed microarray analysis on liver, brain, and adipose tissues from the same mice described above. Despite the lack of broad differential fatty acid composition of the tissues, a high proportion of genes associated with glycolysis and lipid synthesis were upregulated specifically in adipose tissue of *Acss2*^−/−^ mice. Similar changes were not observed in either liver or brain, highlighting the tissue-specific effects of Acss2 deletion. In adipose tissue, the lipogenic genes with increased expression included ATP citrate lyase (Acly), fatty acid synthase (Fasn) and fatty acid elongase 6 (Elovl6). In fact, Elovl6 (log2 fold change +3.59) and Acly (log2 fold change +2.59) were the two most upregulated genes in adipose tissue. The biochemical relationships between the upregulated genes associated with glycolysis and lipogenesis in adipose tissue are shown in Figure 1. Loss of Acss2 acted to upregulate expression of all of these genes, possibly indicating that these changes acted to compensate for the lack of Acss2 activity. Gene upregulation was observed for glycolytic enzymes as well as enzymes of the glycerol-phosphate shunt, which assists the malate-aspartate mitochondrial shuttle in moving reducing equivalents into the mitochondrial matrix. A mitochondrial pyruvate transporter was upregulated, and a catalytic unit of the pyruvate dehydrogenase complex, which acts to convert intramitochondrial pyruvate to acetyl CoA, was also upregulated. A malate transporter that assists the main mitochondrial malate-citrate antiporter was upregulated, along with ATP citrate lyase, which converts citrate into acetyl CoA in the cytoplasm and nucleus. Upregulation was also seen in key enzymes for fatty acid synthesis, including fatty acid synthase, a fatty acid elongase, and two fatty acid desaturases. Overall, these results suggest increased throughput from glucose to unsaturated fatty acids in adipose tissue of *Acss2*^−/−^ mice relative to wild-type mice.

This increased expression pattern for lipogenic genes in adipose tissue could reflect a compensation for the loss of Acss2 activity in *Acss2*^−/−^ mice, and yet similar changes were not observed in the liver, which is a major site of fatty acid synthesis. In contrast to adipose tissue, Huang et al. also found no expression changes in fatty acid associated genes in the livers of *Acss2*^−/−^ mice fed a normal chow diet [13]. Interestingly, even though the expression of lipogenic genes was not significantly affected in the brain, two key fatty acids were lower in the brain: stearic acid (18:0) and oleic acid (18:1 cis9). Genes for enzymes involved in the synthesis of stearic and oleic acids were upregulated in adipose tissue (Figure 1), but those genes were not altered in the brain, so the cause of the lower levels of these two fatty acids in the brain remains to be determined. Because Acly was strongly upregulated in adipose tissue of *Acss2*^−/−^ mice, but not in the liver, it is likely that Acss2 has more a pronounced lipogenic role in adipocytes than in hepatocytes. These results are in agreement with the findings of Huang et al. [13].

### 2.3. Acss2 Loss Institutes Organ-Specific Changes in the Transcriptome and Canonical Signaling Pathways in Liver, Brain and Adipose Tissue

Loss of Acss2 in normally fed mice was associated with 1615 unique, statistically significant DEGs in the three organ systems (*p*-value < 0.05, log2 fold change > 1, z-score > 2). We found 470 significant DEGs in liver, 336 in brain and 809 in adipose tissue (Figure 2A). Out of 470 significant DEGs in the liver, 332 transcripts (70.64%) were downregulated and 138 (29.36%) were upregulated in response to Acss2 deletion. Among 336 DEGs observed in brain, 90 (26.79%) were downregulated and 246 (73.21%) were upregulated in *Acss2*^−/−^ mice. Out of 809 DEGs in adipose, 530 (65.51%) were downregulated and 279 (34.49%) were upregulated in *Acss2*^−/−^ mice. The transcripts with the largest differential expression are shown in Figure 2B–D. Only 19 significant DEGs were common to all three organ systems examined, strongly indicating that Acss2 has very tissue-specific functions. In the brain, almost three quarters of the DEGs were upregulated, whereas in the liver and adipose tissue, approximately two thirds of the DEGs were downregulated. 

To further investigate the system-wide functional significance of the transcriptomic changes seen in these three organ systems, we applied in silico analyses using Ingenuity Pathway Analysis (IPA) and performed core analysis. The obtained DEGs in *Acss2*^−/−^ compared with WT mice were overlaid with the global molecular network in the Ingenuity Pathway Knowledge Base. Comparing *Acss2*^−/−^ to WT mice revealed the association of 39 statistically significant (−log (*p*-value) > 1.3) canonical signaling pathways in the liver, 151 in the brain and 91 in adipose (Figure 2E). There was minimal overlap between altered signaling pathways in the organ systems studied, with only eight canonical signaling pathways common to all three. It is notable that the liver showed the lowest number of significantly altered signaling pathways in response to loss of Acss2. A heatmap of the affected canonical signal transduction pathways (−log (*p*-value) in response to the loss of Acss2 function among the three organ systems is shown in Figure 2F. Significantly altered signaling pathways, associated diseases, and gene networks for each organ system are discussed briefly below. IPA canonical signaling pathways, disease, and cellular function and regulatory biological relationships are highlighted in the text using quotation marks.

### 2.4. Liver

We analyzed activated and inhibited canonical signal transduction pathways in the liver associated with the loss of Acss2 using IPA. The analysis revealed numerous differentially upregulated and downregulated canonical signaling pathways responsive to Acss2 loss. The top 10 statistically significant (−log (*p*-value) > 1.3) and differentially regulated canonical pathways are shown in Figure 3A. Among them, “eNOS signaling”, “ferroptosis signaling pathway”, “LXR/RXR activation”, “xenobiotic metabolism PXR signaling pathway”, and “AMPK signaling” were activated in response to loss of Acss2 function. Downregulated signaling pathways in the liver of *Acss2*^−/−^ mice included “aldosterone signaling in epithelial cells”, “nitric oxide signaling in the cardiovascular system”, “nucleotide excision repair enhanced pathway”, “kinetochore metaphase signaling pathway”, and “the role of MAPK signaling in promoting the pathogenesis of influenza” (Figure 3A). The −log (*p*-values) and z-scores of these canonical signaling pathways are given in Appendix A.

Further, we performed disease and cellular function analysis using IPA to determine upstream regulator, cellular processes, and biological functions in the liver that are modulated by deletion of Acss2. Our analysis showed that Acss2 loss activated upstream regulators Acox1, miR-196a-3p (miRNAs w/seed GGCAACA), and miR-3473h-5p (and other miRNAs w/seed AGGGGCU), leading to alterations in the expression of several genes (listed in Appendix A) associated with disease and cellular functions linked to activation of “anemia” and “organ degeneration” (consistency score = 1.897) (Figure 2C). Similarly, the activation of upstream regulator miR-892b (miRNAs w/seed ACUGGCU) and inhibition of upstream regulator HIF1A, which lead to the altered expression of several genes (listed in Appendix A), was associated with activation of “growth failure” (consistency score = 1.633) (Figure 3C). Further analyses of the DEGs in liver tissue from *Acss2*^−/−^ mice indicated possible inhibition of upstream regulator Lh and activation of upstream regulator miR-34a-5p (and other miRNAs w/seed GGCAGUG), as inferred by alterations in the expression of several genes (listed in Appendix A), and these were associated with activation of “organismal death” (consistency score = −8.667) and inhibition of “infection by RNA virus” (consistency score = −5.367) respectively (Figure 3D,E).

We further analyzed DEGs to determine regulatory biological relationships mediated by the Acss2 in the liver. The top two enriched networks were “amino acid metabolism, carbohydrate metabolism, small molecule biochemistry” (Network 1: score = 62, focus molecules = 33) (Figure 3F), and “cellular development, cellular growth and proliferation, connective tissue development and function” (Network 2: score = 59, focus molecules = 32) (Figure 3G). The corresponding genes and their expression are listed in Appendix A. Overall, these findings indicate regulatory roles for Acss2 in the liver, but do not support a strong role for Acss2 in lipid synthesis or processing in the liver under well-fed conditions. Further, we did observe decreased expression of two enzymes associated with protein and metabolite methylation, namely nicotinamide N-methyltransferase (Nnmt) and adenosylhomocysteinase (Ahcy) (Figure 2B). S-adenosylmethionine (SAM) is the sole methyl donor for SAM-dependent methyltransferases (SDMs), and Nnmt competes with SDMs for the availability of SAM. Decreased Nnmt expression can increase SAM availability for protein and DNA methylation reactions. SDMs transfer the methyl group from SAM to protein and DNA targets, generating S-adenosylhomocysteine (SAH) in the process. Increased concentrations of SAH act to inhibit SDMs. Therefore, it is important to catabolize SAH, which is accomplished by Ahcy. These results suggest that Acss2 may have some regulatory actions over protein and metabolite methylation in the liver.

### 2.5. Brain

Several canonical signaling pathways were modulated by loss of Acss2 in the brain. The top 10 statistically significant (−log (*p*-value) > 1.3) and differentially regulated canonical pathways are shown in Figure 4A. Signaling pathways in the brain that were increased in response to loss of Acss2 function were the “sirtuin signaling pathway”, “3-phosphoinositide biosynthesis”, “superpathway of inositol phosphate compounds”, “Fcγ receptor-mediated phagocytosis in macrophages and monocytes”, “neuropathic pain signaling in dorsal horn neurons”, “D-myo-inositol (1,4,5,6)-tetrakisphosphate biosynthesis in neurons”, and “D-myo-inositol (3,4,5,6)-tetrakisphosphate biosynthesis”. Downregulated signaling pathways in the brain included “oxidative phosphorylation”, “PI3K/AKT signaling”, “BEX2 signaling pathway”, and “cyclins and cell cycle regulation”. The −log (*p*-values) and z-scores of these canonical signaling pathways are given in Appendix A. Several upstream regulator miRNAs in the brain were predicted to be reduced with the loss of Acss2 including miR-1587 (and other miRNAs w/seed UGGGCUG), miR-4687-5p (miRNAs w/seed AGCCCUC), and miR-6777-5p (and other miRNAs w/seed CGGGGAG). These miRNA alter the expression of several genes (listed in Appendix A) associated with disease and cellular functions, including activation of “cell viability of tumor cell lines” and “size of body” (consistency score = 4.111) (Figure 4B).

A number of upstream regulating systems were affected, including “inhibition of the insulin receptor INSR” (z-score = −2.8), as inferred by alterations in the expression of several downstream genes (listed in Appendix A) potentially associating Acss2 with “activation of DNA endogenous promoter” (consistency score = 0.447) (Figure 4C). IPA analysis also indicated activated upstream regulators BDNF and MKNK1, and inhibited upstream regulator miR-4651 (and other miRNAs w/seed GGGGUGG), based on alterations in the expression of several genes (listed in Appendix A). These expression changes are linked to the activation of “cell viability” and “organization of cytoskeleton” (consistency score = 3.395) (Figure 4D). Further analyses in brain tissue from Acss2 deficient mice demonstrated suppression of the upstream regulator miR-6854-3p (miRNAs w/seed GCGUUUC), associated with alterations in the expression of several genes (listed in Appendix A) linked to “inhibition of organismal death” (consistency score = 2) (Figure 4E).

Subsequently, using IPA network analysis, we found the top two enriched networks in the brain were (1) “connective tissue disorders, developmental disorder, hereditary disorder” (Network 1: score = 53, focus molecules = 28), and (2) “metabolic disease, neurological disease, organismal injury and abnormalities” (Network 2: score = 50, focus molecules = 27) (Figure 4F,G). The corresponding genes and their expression changes are listed in Appendix A. Overall, these findings support the conclusion that Acss2 has regulatory actions on numerous signaling pathways in the brain, many of which are unrelated to fatty acid synthesis.

### 2.6. Adipose

In adipose tissue, we analyzed activated and inhibited canonical signal transduction pathways associated with the loss of Acss2. The data indicated numerous differentially regulated (activated or inhibited) canonical signaling pathways that were modulated by Acss2. The top 10 statistically significant (−log (*p*-value) > 1.3) and differentially regulated canonical pathways are shown in Figure 5A. Among them, “oxidative phosphorylation” and “glycolysis I” were activated in response to loss of Acss2 function in adipose tissue. In contrast, the inhibited canonical signaling pathways in adipose tissue of *Acss2*^−/−^ mice included the “sirtuin signaling pathway”, “toll-like receptor signaling”, “iNOS signaling”, “LPS/IL-1 mediated inhibition of RXR function”, “bupropion degradation”, “spliceosomal cycle regulation of the epithelial mesenchymal transition by growth factors pathway”, and “acetone degradation” (Figure 5A). The −log (*p*-values) and z-scores of these canonical signaling pathways are given in Appendix A. It is noteworthy that the sirtuin signaling pathway was inhibited as sirtuins act to regulate Acss2 activity through acetylation [26]. 

IPA disease and cellular function analysis predicted activated upstream regulator PPARG, as inferred by alterations in the expression of several genes (listed in Appendix A) associated with “inhibition of vasculogenesis” (consistency score = 2.739) (Figure 5B). Similarly, activation of upstream regulators miR-149-3p (and other miRNAs w/seed GGGAGGG), miR-486-3p (and other miRNAs w/seed GGGGCAG), and miR-92a-2-5p (miRNAs w/seed GGUGGGG), inferred by alterations in the expression of several genes (listed in Appendix A), were associated with “inhibited infection” of several different cell lines (consistency score = 13.333) (Figure 5C). Additionally, we found activation of upstream regulators Cebpa, Esrra, Hnf1a, IL15, Pcgem1, Psme3, and Slc2a3, as well as inhibition of upstream regulator Clpp linked to alterations in the expression of several genes (listed in Appendix A) associated with “activation of glycolysis” (consistency score = 4.906) (Figure 5D).

Using IPA network analysis, we found the top enriched networks in adipose tissue were “cellular assembly and organization”, “RNA damage and repair”, “RNA post-transcriptional modification” (Network 1, score = 60, focus molecules = 35), and “connective tissue disorders, developmental disorder, hereditary disorder” (Network 2, score = 44, focus molecules = 29) associated with the loss of Acss2 activity (Figure 5E,F respectively). The corresponding genes and their expression changes are listed in Appendix A. It is noteworthy that the most enriched network in adipose tissue included RNA post-transcriptional modification, indicating that Acss2 has regulatory actions at both the pre- and post-transcriptional levels.

### 2.7. g:Profiler Functional Enrichment Analysis

Enrichment analysis with gProfiler2 was used to map the effects of Acss2 deletion on the over-representation of bioprocesses (GO:BP) and transcription factor motifs (TF) in the three tissues. We found tissue-specific patterns of functional enrichment for the statistically significant biological processes and TF motifs. Figure 6 shows Manhattan plots generated for upregulated and downregulated DEGs in each tissue. Manhattan plots are used to visualize functional enrichment (over-representation), with the negative log10 of the adjusted *p* values plotted on the *Y* axis, and functional terms arranged as colored dots on the *X* axis. The tissue-specific patterns of affected functions were particularly evident among TF in the three organ systems. In the liver and brain, for example, the pattern of TF enrichment was the opposite, with strong over-representation of TF among downregulated DEGs in the liver (Figure 6B), as contrasted with strong over-representation of TF in the upregulated genes in the brain (Figure 6C). These findings further substantiate the tissue-specific regulatory roles of Acss2 and raise the possibility that Acss2 may act primarily as a transcriptional repressor in the brain, and a transcriptional activator in the liver, while having a mixed role in the regulation of transcription factor expression in adipose tissue. The dichotomous effect of Acss2 deletion on TF expression in the brain and liver warrants further attention.

### 2.8. Karyoplot Analysis

Karyoplots were constructed to show the chromosomal distribution of DEGs in liver, brain and adipose tissue. Upregulated and downregulated DEGs were plotted separately for each tissue (Appendix A). Loss of Acss2 resulted in widespread expression changes distributed in a tissue-specific manner across all autosomes. A modest number of expression changes were also observed on X chromosomes in all three organ systems. In contrast, no expression changes were seen on the Y chromosome in any tissue.

## 3. Discussion

We examined the effects of Acss2 deletion in three organ systems with distinct lipid metabolism behaviors. The liver is involved in many aspects of lipid physiology, including being a major site of lipid and lipoprotein synthesis, processing, and release. The brain is a site of complex lipid metabolism required to generate and maintain the myelin sheathing of neuronal axons and support the membrane lipid turnover necessary for efficient neurotransmission. Adipose tissue is associated with lipid synthesis, deposition, storage, and release, depending on nutritional availability. The results we obtained in response to Acss2 deletion in these three organ systems demonstrate a strong effect on cellular signaling pathways, but relatively minimal effects on fatty acid content. In adipose tissue, a number of genes associated with glycolysis and fatty acid synthesis were upregulated in *Acss2*^−/−^ mice (Figure 1), but among fatty acids, only myristic acid was elevated in adipose tissue. Huang and colleagues studied lipid metabolism in *Acss2*^−/−^ mice and found that loss of Acss2 had opposing effects under fed and fasted conditions [13]. They noted that Acss2 acted in a transcription factor-like way to promote the storage of fats when food was abundant, but in turn Acss2 enhances the metabolism of fats during fasting through the selective regulation of genes involved in lipid metabolism. These findings are in agreement with our results showing that Acss2 deletion has profound effects on transcription factor expression in all three organ systems examined (Figure 6).

In general, the effects of Acss2 deletion on signaling pathways were intriguingly specific for each organ system, with relatively limited overlap. However, in brain and adipose tissue, two canonical signaling pathways were strongly perturbed, including “sirtuin signaling pathway” and “oxidative phosphorylation” (Figure 4A and Figure 5A). However, the effects of Acss2 deletion in the two tissues were opposite, with the sirtuin pathway being activated in the brain, but inhibited in adipose tissue. Oxidative phosphorylation, on the other hand, was inhibited in the brain but was stimulated in adipose tissue. This indicates that the regulatory effects of Acss2 reflect the physiological roles of different organ systems. Adipose tissue is a net exporter of energy substrates, especially when food is limiting, whereas the brain is a net consumer, and Acss2 clearly plays complementary roles in the regulation of energy distribution and utilization in these tissues. Sirtuins are NAD+ dependent deacetylases that regulate metabolism. For example, sirtuin-1 (Sirt1) acts to deacetylate and thereby activate Acss2, and to increase fatty acid synthesis from acetate [26]. The effects of Acss2 deletion on signaling pathways are consistent with the observation that Acss2 can translocate to the cell nucleus where it can regulate transcription [13,18,19,27]. Earlier studies in the brain demonstrated a predominant localization of Acss2 in the nuclei of neurons, astrocytes and oligodendrocytes, with greatly increased expression 24 h following traumatic brain injury [16]. The substantially increased nuclear expression of Acss2 after brain injury indicates that the enzyme is likely involved in stress responses and possibly tissue repair.

While the gene expression differences in the current study were mostly distinct for each organ system, it is noteworthy that we observed large expression changes for transmembrane protein-25 (Tmem25) in the liver and the brain. Tmem25 was the most downregulated gene observed in both tissues (Figure 2B,C). Tmem25 has been associated with lysosomal acidification and accelerated protein degradation in neurons [28]. Because the loss of Acss2 substantially reduced the expression of Tmem25 in both the liver and brain, it is possible that Acss2 acts to enhance Tmem25 transcription during autophagic responses to increase lysosomal acidification and facilitate protein turnover. Further, proteasome 26S subunit, non-ATPase-8 (Psmd8) was downregulated in all three tissues (Figure 2B–D). Psmd8 is part of the 19S regulatory proteasome subunit. It is therefore possible that Acss2 upregulates the expression of proteasome subunits during autophagic responses. Our results are in agreement with previous studies on Acss2 and autophagy [18,19,29].

### Acss2 as an Epigenetic Regulator and Metabolic Integrator

Chromatin can assume multiple distinct configurations that modulate gene expression by making transcription sites accessible or inaccessible. Specific chromatin states induced by post-translational modifications of nucleosome proteins are brought about in part by enzymes that also play roles in energy metabolism [1,2]. The role of metabolic enzymes acting in the cell nucleus is an emerging theme in transcriptional regulation. Acss2 acts as one of these epigenetic regulatory enzymes involved in chromatin remodeling that also plays a role in central acetyl-CoA metabolism (reviewed in [24]). Acss2 regulates the transcription of specific gene suites by at least two mechanisms, the acetylation of histones acting to open specific nucleosome sites, and by the acetylation of specific transcription factors (reviewed in [25]). By interacting with transcription factor complexes, Acss2 provides acetyl-CoA on-demand to histone acetyltransferases (HAT) that utilize acetyl-CoA to open chromatin and allow access of transcriptional enzyme complexes. A third way for Acss2 to facilitate transcription is the local production of AMP near nascent transcription sites, which could increase nuclear AMPK activity to phosphorylate and activate targets such as histone acetyltransferase-1 [30]. Acss2 also interacts with p300 and CREB binding protein (CBP), which are transcriptional coactivators with acetyltransferase activity that target histones and other transcription-associated proteins for acetylation [31]. In this role Acss2 provides on-demand acetyl-CoA for the activity of p300 and CBP. For example, in certain cancer cells exposed either to hypoxia or glucose deprivation, hypoxia inducible factor 2-alpha (HIF-2α) binds to an erythropoietin promoter region and enhances erythropoietin expression. HIF-2α acetylation requires Acss2 and CBP, and Acss2 is required for stable CBP/HIF-2α complex formation [17]. In this manner, HIF-2α acetylation by the sequential activity of Acss2 and CBP improves the response to low oxygen tension. Further, Acss2 expression is increased in the nuclei of differentiating neurons [16] and localizes to chromatin regions of increased histone acetylation and transcriptional activity [32]. Acetate, acting through Acss2, activates transcriptional programs in the brain linked to learning and memory, and loss of Acss2 reduces nuclear acetyl-CoA levels, histone acetylation, and expression of associated genes [32,33]. These investigators examined transcription factor binding motifs for Acss2 and found an association with the acetyltransferases p300 and CBP, as in the case of enhanced erythropoietin expression noted above. 

Autophagy is associated with nutrient deprivation and cellular stress responses, and here again Acss2 plays a regulatory role. AMPK activation promotes autophagy, and we found AMPK signaling to be perturbed in all three organ systems examined in *Acss2*^−/−^ mice (Figure 2F). Nutrient deprivation leads to AMPK activation, which in turn phosphorylates Acss2, resulting in Acss2 translocation to the cell nucleus. In conjunction with transcription factor EB (Tfeb), Acss2 associates with lysosomal and autophagosomal gene promoter regions, and utilizes acetate derived from histone deacetylation to locally produce acetyl-CoA for acetylation of histone H3, thereby promoting autophagic gene expression [18,19]. As such, Acss2 is at the hub of interactions between acetyl-CoA metabolism and transcriptional regulation of genes associated with numerous cellular functions ranging from lipid metabolism to autophagy to memory formation. In this capacity Acss2 acts to augment or enhance distinct regimens of transcriptional activity in different tissues during development and under various stressors, including nutrient deprivation and injury. 

Our current results support the conclusion that Acss2, while predominantly redundant to Acly in lipid synthesis, is unique in its metabolic regulation and signaling roles. The observed changes occurred in well-fed, non-stressed mice, and therefore reflect only a subset of expression changes that would be expected under stressed conditions or in response to injury. Our results can be explained by Acss2 acting as a transcriptional regulator through selective on-demand acetyl-CoA synthesis at sites of enhanced transcriptional activity. As such, Acss2 provides acetyl-CoA not only for histone acetylation, but also for acetylation of transcription factors and associated protein targets that act to modulate transcription. Because the differential expression patterns mapped to transcription factors in an organ-specific manner, Acss2 binds to various transcription factor complexes and assists in regulating the transcriptional expression of specific genes during various cellular stress responses. It is very likely that the organ-specific nature of Acss2 regulatory action will also translate to species-specific differences in translational modulation. This is especially true, e.g., with ruminants, where reliance on acetate production from gut microbiota is a key source of nutrition [24]. It can be anticipated that Acss2 will have distinct modulatory actions in different species based on the specifics of their modes of nutrient acquisition and physiology. In conclusion, Acss2 can be considered a task-switching metabolic effector that transitions from roles in cytoplasmic acetyl-CoA formation and lipid synthesis to become a nuclear transcriptional regulator through its targeted actions on histone and transcription factor acetylation and other actions in the cell nucleus, e.g., locally increased levels of AMP at sites of high Acss2 activity. Future research on the underlying mechanisms whereby Acss2 acts to institute various metabolic programs in different organ systems will shed light on the pathophysiology of diseases ranging from metabolic disorder to cancer. 

## 4. Materials and Methods

### 4.1. Acss2 Gene Knockout

Breeding pairs of Acss2 gene (NM-019811) knockout mice were obtained from Lexicon Pharmaceuticals (Bridgewater, NJ, USA) via Taconic Farms (Germantown, NY, USA) and they were bred in our animal facility on a C57BL background. The Acss2 gene knockout involved standard procedures using embryonic stem cells with the 129S5 genetic background. The deleted region included approximately 3.6 kb beginning at the transcription start site. Deletion of Acss2 was confirmed using RT-PCR (Transnetyx, Inc., Cordova, TN, USA). 

### 4.2. Animal Breeding

Animal care and experimental procedures were carried out in accordance with NIH guidelines and approved by the Uniformed Services University Animal Care and Use Committee. Animals were housed in an environmentally controlled room (20–23 °C, ~44% humidity, 12 h light/dark cycle, 350–400 lux, lights on at 6:00 am), with food and water available continuously. Animal handling was minimized to reduce animal stress. The homozygous *Acss2^−/−^* mice reproduced normally, and therefore wild-type mice and homozygous mice were bred separately for the experiments. Before use in experiments, individual genotypes were confirmed via PCR using probes designed by Lexicon Pharmaceuticals.

### 4.3. Fatty Acid Analysis

For tissue fatty acid determinations, 5 male mice were used per group. Liver, brain, and mesenteric fat were rapidly collected, immediately homogenized in Trizol and the homogenates were frozen rapidly on dry ice. Tissue homogenates were thawed, and 10 μL aliquots were mixed with 50 μL of 1M NaOH and left for 60 min at room temperature in the dark. The mixtures were then acidified with 1 M HCl to pH 3–4, 10 ng of fatty acid internal standard mix was added, and the solutions were saturated with NaCl. The mixtures were extracted with isooctane-ethyl acetate (9:1) four times and the extracts from each sample were pooled. Samples were then dried and dissolved in 65 μL of solvent B (methanol-water-ammonium acetate; 95:5:0.1%, pH 7.6). To each sample, 35 μL of solvent A was added (methanol-acetonitrile-water-ammonium acetate; 5:5:85:0.1%, pH 7.6). An aminopropyl-Strata column was conditioned with di-isopropyl ether followed by hexane, and the samples loaded. Columns were eluted with 3 mL di-isopropyl ether-formic acid (98:2). Eluates were dried and dissolved in a mixture of solvents A and B (35:65%). LC-MS was performed as previously described [34].

### 4.4. Gene Array

Gene array analysis was performed on tissues from normally fed (non-fasted) adult male *Acss2*^−/−^ and age-matched wild-type mice of the same strain (15–16 weeks old). Liver, brain, and mesenteric fat were rapidly collected, immediately homogenized in Trizol, and the homogenates were frozen rapidly on dry ice. RNA was extracted, frozen, and processed at the University of Chicago Genomics Facility. RNA quality and quantity were checked using an Agilent Bioanalyzer (Santa Clara, CA, USA). RNA was processed into biotinylated cRNA using the Ambion Illumina^®^ TotalPrep™ RNA Amplification Kit (ThermoFisher, Waltham MA, USA) and the biotinylated cRNA was hybridized to Illumina microarray (San Diego CA, USA) using an Illumina provided protocol. Staining of the arrays and scanning on an Illumina HiScan were also performed using Illumina provided protocols. Tissues examined included liver (*n* = 5 *Acss2*^−/−^ and 4 wild-type), brain (*n* = 3 *Acss2*^−/−^ and 3 wild-type), and adipose tissue (*n* = 4 *Acss2*^−/−^ and 3 wild-type).

### 4.5. Gene Alignment and Counts

Reads were aligned to the *Mus musculus* genome version mm10 [35] using Star Aligner [36] and the RSEM software package to quantify transcripts [37]. Resulting TPM/FPKM counts were used in the differential expression analysis using the limma R software package [38]. Samples were grouped according to treatment and tissue type. DEG lists were arranged according to the foldchange and associated *p*-value. These DEG lists were used for further enrichment analysis as described below. 

### 4.6. Pathway Analyses

In silico analyses of DEGs, (*p*-value < 0.05, log2 fold change > 1) in *Acss2*^−/−^ mice compared to wild-type control groups in the liver, brain, and mesenteric adipose tissue were performed using Qiagen’s Ingenuity Pathway Analyses system (IPA, QIAGEN Inc., Germantown MD, USA, https://digitalinsights.qiagen.com/ingenuity-pathway-analysis-resources/, accessed during December 2022). The analyses were based on experimentally observed and predicted data from the Ingenuity Knowledge Base data sources using the Benjamini–Hochberg corrected right-tailed Fisher’s exact test.

### 4.7. g:Profiler Enrichment Analysis

Functional enrichment analysis of the DEG in the 3 tissues was performed with g:Profiler2 (https://biit.cs.ut.ee/gprofiler/page/r, accessed on 4 January 2023) using the gost command with native options (https://biit.cs.ut.ee/gprofiler/gost). Manhattan plots were generated using the gostplot function. Manhattan plots depict the over-representation of negative log10 adjusted *p* values [−log10 (p adj)] on the *Y*-axis, and clustered groups of functional terms as columns along the *X*-axis. Queries included gene ontology (GO) biological processes, (GO:BP), molecular functions (GO:MF), cellular components (GO:CC), as well as KEGG pathways (K), Reactome pathways (R), and TransFac transcription factor binding motifs (TF).

### 4.8. Karyoplot Analysis

Karyoplots were done using KaryoploteR in the R programming language [39]. DEG lists were used to map the expression patterns onto mouse chromosomal positions. All statistically significant DEGs (*p* < 0.05) were included. The log fold change was mapped to the size of the dots on the karyoplots.

## Figures and Tables

**Figure 1 ijms-24-03673-f001:**
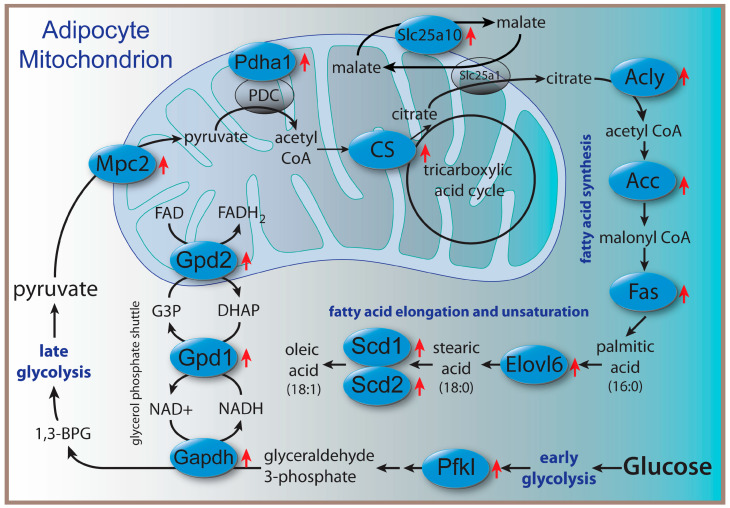
Schematic of major gene expression increases observed in adipose tissue associated with glucose utilization for lipid synthesis in *Acss2*^−/−^ mice. Proteins whose genes showed significant upregulation are indicated by red arrows. Despite the increased expression of lipogenic genes, only myristic acid (C14:0) was increased in abundance in adipose tissue. Abbreviations: 1,3-BPG = 1,3-bisphosphoglycerate, ACC = acetyl CoA carboxylase, Acly: ATP citrate lyase, CS = citrate synthase, DHAP = dihydroxyacetone phosphate, Elovl6 = fatty acid elongase 6, Fas = fatty acid synthase, G3P = glycerol 3-phosphate, Gapdh = glyceraldehyde 3-dehydrogenase, Gpd1 and 2 = glycerol 3-phosphate dehydrogenases 1 (cytoplasmic) and 2 (mitochondrial), Mpc2 = mitochondrial pyruvate carrier 2, Pdah1 = pyruvate dehydrogenase E1 alpha 1, PDC = pyruvate dehydrogenase complex, Pfkl = phosphofructokinase, liver, Scd1 and Scd2 = stearoyl-CoA desaturases 1 and 2, Slc25a1 = solute carrier family 25, member 1 (citrate carrier), Slc25a10 = solute carrier family 25, member 10.

**Figure 2 ijms-24-03673-f002:**
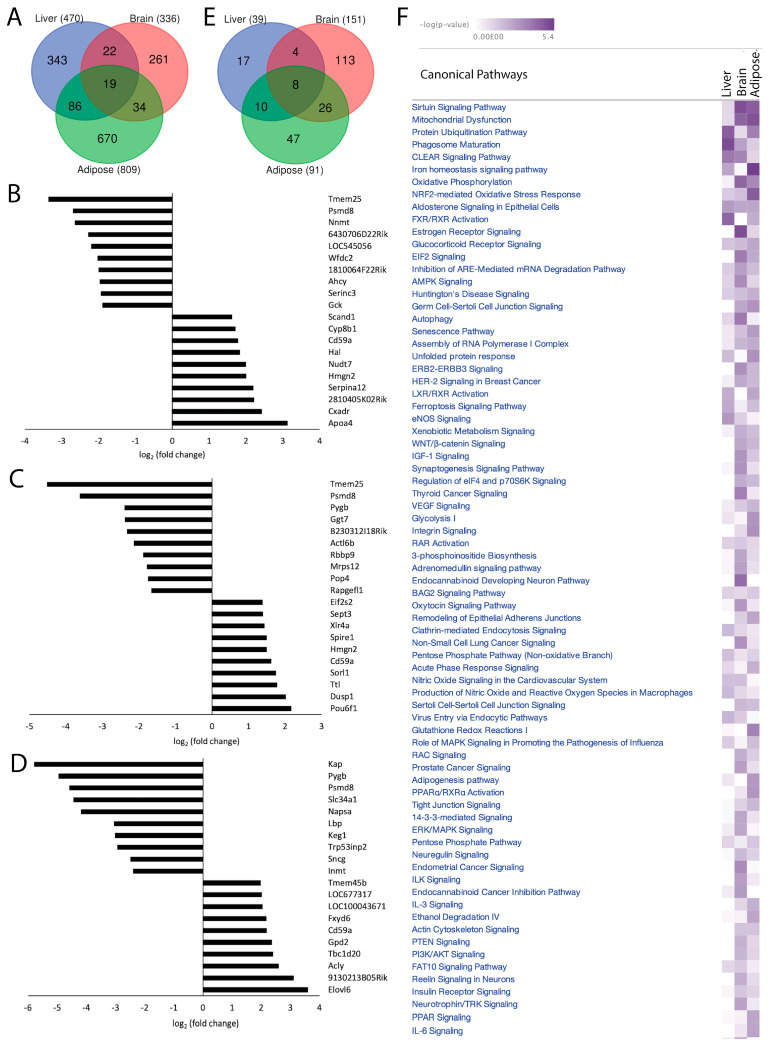
Acss2 loss institutes tissue-specific changes in the transcriptome and canonical signaling pathways in mouse liver, brain and adipose tissue. DEGs were determined in *Acss2*^−/−^ mice, as compared with wild-type control groups, in three metabolically disparate organ systems including the liver, brain and mesenteric adipose tissue. (**A**) Venn diagram showing the common and specific statistically significant DEGs (*p*-value < 0.05, log2 (fold change) > 1). The bar graphs show the top 10 downregulated and upregulated genes in (**B**) liver, (**C**) brain, and (**D**) adipose tissue from *Acss2*^−/−^ mice compared to wild type mice. (**E**) Venn diagram depicting total number of common and specific statistically significant canonical signal transduction pathways (−log (*p*-value) > 1.3). (**F**) Heatmap showing the canonical signal transduction pathways that were significantly altered in response to loss of Acss2 function among the 3 tissues (−log (*p*-value)).

**Figure 3 ijms-24-03673-f003:**
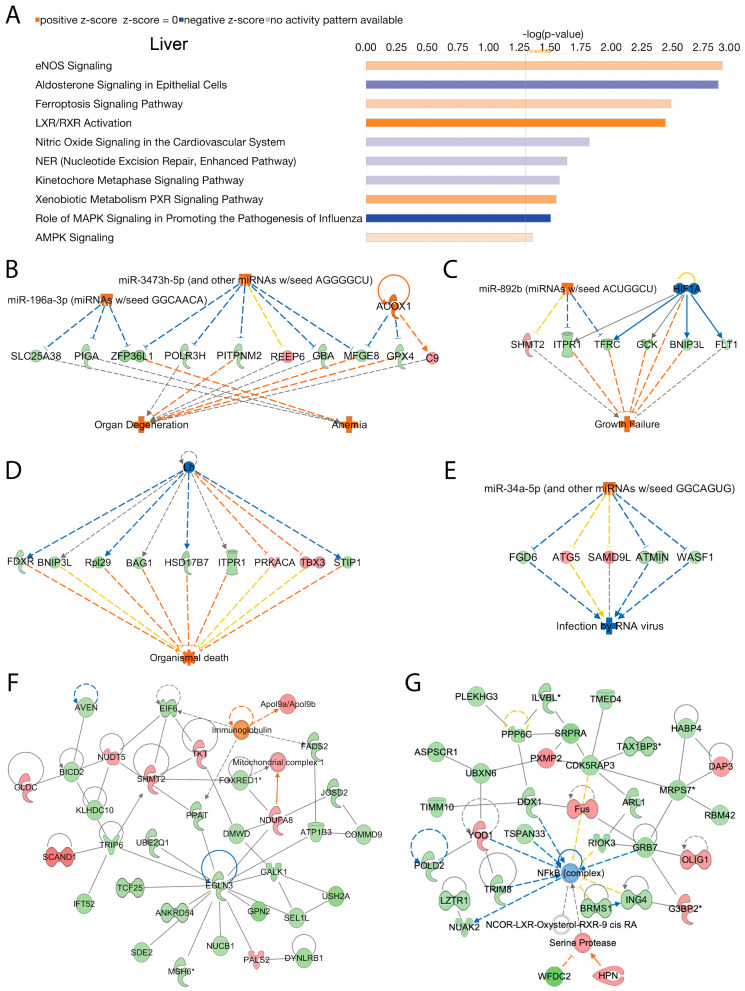
**Liver**; ACSS2 regulates several canonical signaling pathways, upstream transcriptional regulatory molecules, and cellular and molecular networks in murine liver. (**A**) The bar graph shows the top 10 activated or inhibited canonical signal transduction pathways in response to *Acss2* deletion. The bars indicate the −log (*p*-value) of over-representation, while the orange line indicates the threshold. The color of the bars indicates the sign of the z-score (orange for positive z-score, blue for negative). The diagrams show the most enriched upstream transcriptional regulatory molecules including; (**B**) Acox1, miR-196a-3p and miR-3473h-5p; (**C**) HIF1A and miR-892b; (**D**) Lh and (**E**) miR-34a-5p, that are associated with disease and cellular function associated with the differences in gene expression between *Acss2*^−/−^ and wild type mice. The color scheme indicates the value of the differential expressions and the shapes indicate the type and function, as indicated in Appendix A. Solid arrows represent the genes that interact directly, whereas dotted arrows represent indirect interactions between genes. The most enriched networks included (**F**) Network 1: “Amino acid metabolism, carbohydrate metabolism, small molecule biochemistry”, and (**G**) Network 2: “Cellular development, cellular growth and proliferation, connective tissue development and function”, which were associated with the differences in gene expression between *Acss2*^−/−^ (*n* = 5) and wild-type mice (*n* = 4).

**Figure 4 ijms-24-03673-f004:**
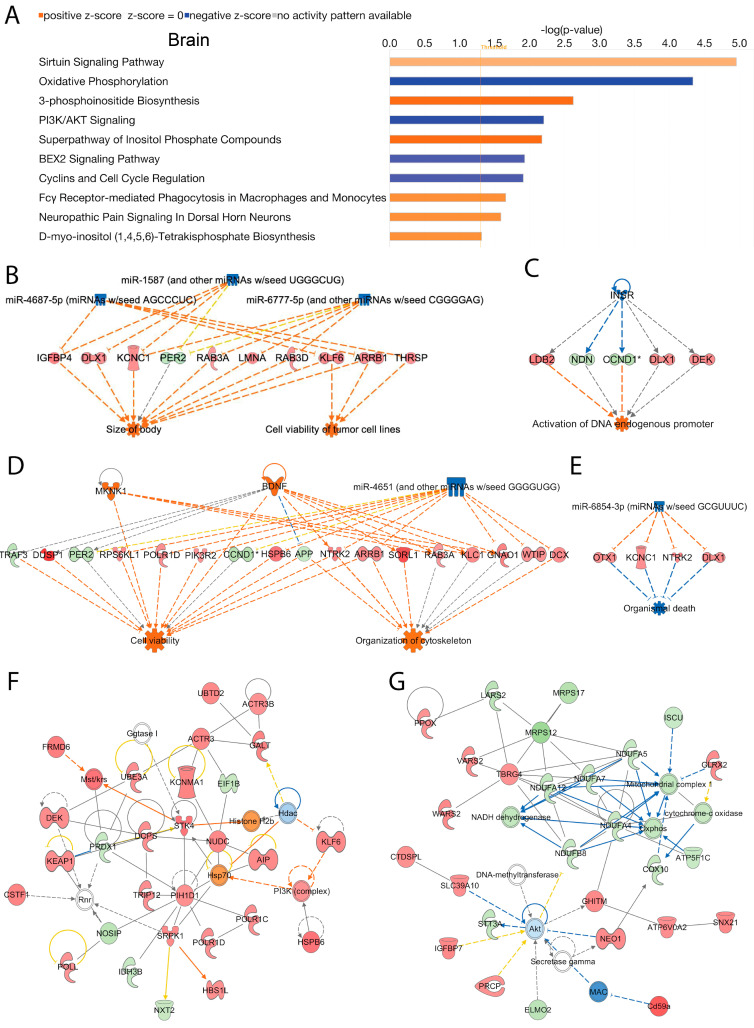
**Brain**; Acss2 regulates several canonical signaling pathways, upstream transcriptional regulatory molecules, and cellular and molecular networks in murine brain. DEGs in *Acss2*^−/−^ mice compared to wild-type mice in brain tissue were analyzed using IPA. (**A**) The bar graph shows the top 10 activated or inhibited canonical signal transduction pathways. The bars indicate the −log (*p*-value) of over-representation, while the orange line indicates the threshold. The color of the bars indicates the sign of the z-score (orange for positive z-score, blue for negative). The diagrams show the most enriched upstream transcriptional regulatory molecules including; (**B**) miR-1587, miR-4687-5p and miR-6777-5p; (**C**) INSR; (**D**) BDNF, miR-4651, MKNK1 and (**E**) miR-6854-3p, that are associated with disease and cellular function. The color scheme indicates the value of the differential expressions and the shapes indicate the type and function, as indicated in Appendix A. Solid arrows represent the genes that interact directly, whereas the dotted arrows represent indirect interactions between genes. The most enriched networks included (**F**) Network 1: “Connective tissue disorders, developmental disorder, hereditary disorder”, and (**G**) Network 2: “Metabolic disease, neurological disease, organismal injury and abnormalities”, which were associated with the differences in gene expression between *Acss2*^−/−^ (*n* = 3) and wild-type mice (*n* = 3).

**Figure 5 ijms-24-03673-f005:**
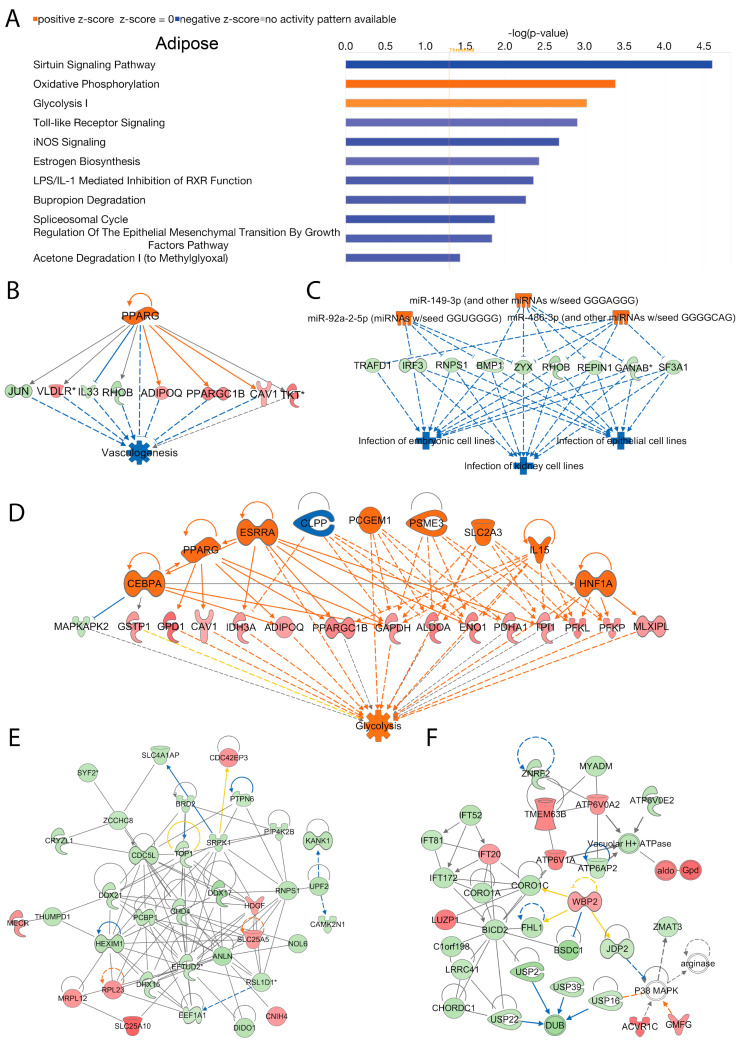
**Adipose**; Acss2 regulates several canonical signaling pathways, upstream transcriptional regulatory molecules, and cellular and molecular networks in murine adipose tissue. DEGs in adipose tissue of *Acss2*^−/−^ mice were compared to those of wild-type mice and were analyzed using IPA. (**A**) The bar graph shows the top 10 activated or inhibited canonical signal transduction pathways. The bars indicate the −log (*p*-value) of over-representation, while the orange line indicates the threshold. The color of the bars indicates the sign of the z-score (orange for positive z-score, blue for negative). The analysis indicated the most enriched upstream transcriptional regulatory molecules including; (**B**) Pparg; (**C**) miR-149-3p, miR-486-3p and miR-92a-2-5p and (**D**) Cebpa, Clpp, Esrra, Hnf1a, IL15, Pcgem1, Pparg, Psme3 and Slc2a3, that are predicted to be associated with disease and cellular functions. The color scheme indicates the value of the differential expressions and the shapes indicate the type and function, as indicated in Appendix A. Solid arrows represent the genes that interact directly, and the dotted arrows represent indirect interactions between genes. The most enriched networks included (**E**) Network 1: “Cellular assembly and organization, RNA damage and repair, RNA post-transcriptional modification”, and (**F**) Network 2: “Connective tissue disorders, developmental disorder, hereditary disorder”, which were associated with differences in gene expression between *Acss2*^−/−^ (*n* = 4) and wild-type mice (*n* = 3).

**Figure 6 ijms-24-03673-f006:**
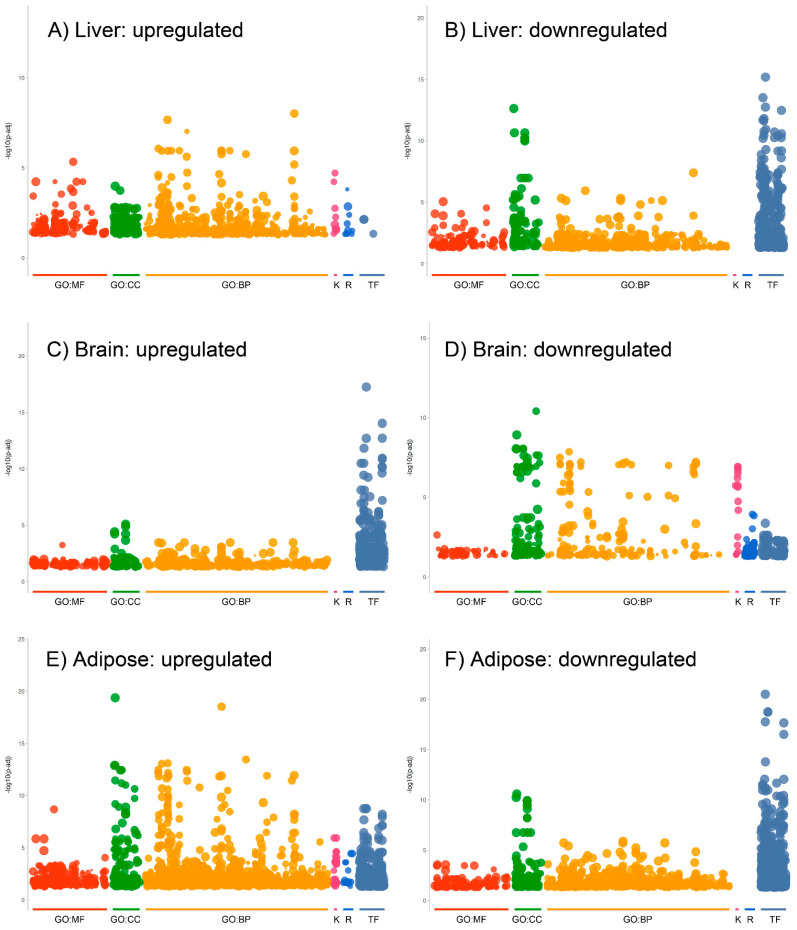
g:Profiler2 Manhattan plots of the organ-specific effects of Acss2 deletion. (**A**) Analysis of upregulated DEG in liver revealed enrichment of 820 biological processes (GO:BP), but indicated over-representation of only 13 transcription factor motifs (TF). (**B**) Downregulated DEGs in liver showed over-representation of 239 GO:BP and 590 TFs. (**C**) Upregulated DEGs in brain showed 711 enriched biological processes and 466 enriched TFs. (**D**) Downregulated genes in brain indicated enrichment of 676 biological processes and enrichment of 98 TF. (**E**) Upregulated genes in adipose tissue indicated over-representation of 1330 GO:BP and 397 TF. (**F**) Downregulated genes in adipose tissue revealed enrichment of 1497 GO:BP and 578 TF. Overall, there were significant effects on transcription factor expression, and in the cases of the brain and liver, the effects were directional (TF mostly upregulated in the brain, but mostly downregulated in the liver). The X *axes* show the grouped functional terms and the Y *axes* give the adjusted enrichment *p* values in negative log10 [−log10(p-adj)]. Abbreviations: GO:BP = biological processes, GO:MF = molecular function, GO:CC = cellular components, K = KEGG pathways, R = Reactome pathways, TF = TransFac transcription factor binding motifs.

## Data Availability

Data are available upon request.

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
