# Peer review of "Acss2 Deletion Reveals Functional Versatility via Tissue-Specific Roles in Transcriptional Regulation"

_ijms, 2023, doi:10.3390/ijms24043673_

Round 1
Reviewer 1 Report
Submission describes specificities in differential expression of hundreds key metabolic enzymes genes and signal transductions factors in livers, adipose tissues and the brains of AccS2(-/- KO and wild type mice. Relatively small changes in final products (FA) concentrations, accompanied by compensatory increases of ACL and acetyl-CoA synthesizing enzymes genes expression, confirm earlier phenotypic observations on the existence of flexible mechanisms maintaining stable levels of this key metabolite in the tested organs. Perhaps changes in Accs2-dependent nuclear acetylations might contribute to observed alterations. The paper provides genome-vise data on Acss2 deletion bearing animals, what is of encyclopedic-like significance. The discussion is missing explanation how this could be related to known phenotypic data on this subject. Hence, the discussion should try to integrate author’s genomic data with metabolic(omic) and enzymological findings on whole organ and cellular level, provided by other authors.
Author Response
Reviewer #1: Submission describes specificities in differential expression of hundreds key metabolic enzymes genes and signal transductions factors in livers, adipose tissues and the brains of AccS2(-/- KO and wild type mice. Relatively small changes in final products (FA) concentrations, accompanied by compensatory increases of ACL and acetyl-CoA synthesizing enzymes genes expression, confirm earlier phenotypic observations on the existence of flexible mechanisms maintaining stable levels of this key metabolite in the tested organs. Perhaps changes in Accs2-dependent nuclear acetylations might contribute to observed alterations. The paper provides genome-vise data on Acss2 deletion bearing animals, what is of encyclopedic-like significance.
Comment 1: The discussion is missing explanation how this could be related to known phenotypic data on this subject. Hence, the discussion should try to integrate author’s genomic data with metabolic(omic) and enzymological findings on whole organ and cellular level, provided by other authors.
Response: We thank the reviewer for their helpful comments and suggestions. Concerning linkages between the differentially expressed genes and known phenotypic changes in Acss2 deficient mice, we did discuss some of the relevant associations with respect to Acly compensation, etc. However, as noted in the manuscript, well-fed Acss2 deficient mice display no obvious phenotypic differences from wild type mice. Much larger differences are noted in fasted Acss2 KO mice (Huang et al., 2018). We are currently investigating the gene expression changes in Acss2 KO mice that have been fasted, as well as those given high doses of acetate in the diet, and these findings will be reported in a future publication.
Based on the reviewer’s comment, we have made additions to the discussion to further integrate our findings with the published literature.
References
Huang, Z., Zhang, M., Plec, A.A., Estill, S.J., Cai, L., Repa, J.J., McKnight, S.L., and Tu, B.P. (2018). ACSS2 promotes systemic fat storage and utilization through selective regulation of genes involved in lipid metabolism. ProcNatlAcadSciUSA 115, E9499-E9506.
Reviewer 2 Report
It is interesting to see that the highly expressed gene Nnmt is low expressed in Acss-/- mutant mice. However, in Figure 2B the top 10 downregulated genes Nnmt is included 2 times. Is it a distinct form of Nnmt?
There are some genes (NNmt, Tmem25 etc.) that were differentially expressed in different tissue types. But did the author look at the expression of genes spatially? It would be interesting if the author can include figures of top differentially expressed genes spatially.
Also, the expression of top differentially expressed genes can be validated by RT-PCR.
The quality of the figures needs to be improved. It was hard to look at the Karyoplot. Other figures can also be uploaded with better quality.
Author Response
Reviewer #2:
Comment 1: It is interesting to see that the highly expressed gene Nnmt is low expressed in Acss-/- mutant mice. However, in Figure 2B the top 10 downregulated genes Nnmt is included 2 times. Is it a distinct form of Nnmt?
Response: We thank the reviewer for their valuable comments and suggestions. In Figure 2B Nnmt was shown twice because two distinct Nnmt probes on the gene chip were significantly different in the knockout mice. We agree with the reviewers’ comments that it appeared redundant and we have therefore removed the second Nnmt from Figure 2B and added the next most altered gene to the list (Gck – glucokinase).
Comment 2: There are some genes (Nnmt, Tmem25 etc.) that were differentially expressed in different tissue types. But did the author look at the expression of genes spatially? It would be interesting if the author can include figures of top differentially expressed genes spatially. Also, the expression of top differentially expressed genes can be validated by RT-PCR.
Response: Indeed, as suggested by the reviewer, spatial DEG assessments followed by RT-PCR confirmation of mRNA expression changes are important analyses to perform. These are unfortunately beyond the scope of this manuscript, but we plan to include them as part of our future goals in ongoing studies.
Comment 3: The quality of the figures needs to be improved. It was hard to look at the Karyoplot. Other figures can also be uploaded with better quality.
Response: Please note that the figure quality in the compiled review version of the manuscript is lower, and higher resolution figures have been uploaded to the journal’s website for the published version. Many of the high-resolution figures are intended to be included in the published version as whole-page figures so that the details and text are easily readable.
Regarding the karyoplot figure, this is the best quality that we have been able to get. Based on the reviewer’s comments we have moved the karyoplot to the online supplemental section, and replaced it with gProfiler Manhattan plots that had been in the supplementary material. The Manhattan plots are more relevant to the discussion on transcription factors and therefore are more appropriate than the karyoplots for the main text.